# Unexpected complexity of everyday manual behaviors

Yuke Yan[1], James M. Goodman[1], Dalton D. Moore[1], Sara A. Solla[2] & Sliman J. Bensmaia [ID] [1,3,4 ✉]

How does the brain control an effector as complex and versatile as the hand? One possibility is that neural control is simplified by limiting the space of hand movements. Indeed, hand kinematics can be largely described within 8 to 10 dimensions. This oft replicated finding has been construed as evidence that hand postures are confined to this subspace. A prediction from this hypothesis is that dimensions outside of this subspace reflect noise. To address this question, we track the hand of human participants as they perform two tasks—grasping and signing in American Sign Language. We apply multiple dimension reduction techniques and replicate the finding that most postural variance falls within a reduced subspace. However, we show that dimensions outside of this subspace are highly structured and task dependent, suggesting they too are under volitional control. We propose that hand control occupies a higher dimensional space than previously considered.

[1] Committee on Computational Neuroscience, University of Chicago, Chicago, IL, USA. [2] Department of Physiology, Northwestern University, Chicago, IL, USA. [3] Department of Organismal Biology and Anatomy, University of Chicago, Chicago, IL, USA. [4] Grossman Institute for Neuroscience, Quantitative Biology, and Human Behavior, University of Chicago, Chicago, IL, USA. ✉email: sliman.bensmaia@gmail.com

From picking up a coffee cup to playing the piano, humans can engage in a wide range of manual behaviors, highlighting the staggering flexibility of the hand, whose 27 bones and 39 muscles give rise to >20 biomechanical degrees of freedom (DOF) for volitional movement[1,2]. This versatility is also supported by a sophisticated neural system: the hand is one of the most densely innervated regions of the human body[3], and the hand representation in sensorimotor cortex is disproportionately large[4]. In addition, although primate motor cortex lacks a clear somatotopic organization[5], it nonetheless seems to contain a specialized module for hand control[6].

The complexity of the hand has called into question whether the central nervous system (CNS) can fluidly control such a complex effector[7,8]. An appealing hypothesis is that hand postures—which in principle can exist over 20 or more DOF—are reduced to a lower-dimensional manifold to simplify the control problem[7–9]. Instead of spanning the full space afforded by all available DOFs, hand control relies on a set of synergies that are combined to give rise to manual behaviors. Broadly defined, a synergy is a set of muscle activations or joint movements that are recruited collectively rather than individually[10]. This restriction of volitional movements of the hand to combinations of a small number of synergies is presumed to confer a number of computational advantages, including robustness to noise and facilitated learning of novel movements[8,9]. To date, many studies have looked for hand synergies in different movement contexts, with a mixture of evidence for and against the notion that the CNS indeed constrains hand movements to the subspace spanned by synergies as a limited basis set[9–18].

The most compelling evidence for hand postural synergies stems from analysis of hand kinematics or the muscle activations that drive them, which seem to occupy a lower-dimensional manifold within the space spanned by the collective DOFs. Indeed, principal component analysis (PCA) of kinematics or muscle activations reveals that a small number of principal components (PCs) account for most of the variance in the movements or activations associated with a given behavior (e.g., grasping, playing piano, and typing)[11,12,19–22]. The tacit assumption underlying the interpretation of this dimensionality reduction is that high-variance PCs—the synergies—are under volitional control, whereas low-variance PCs reflect motor or measurement noise. Another possibility, however, is that the exquisite control of the hand is mediated by high-dimensional sensorimotor signals, and that low-variance PCs are critical to achieving precise hand postures. A previous investigation of this question concluded that grasping movements were confined to six dimensions or less[22]—still far fewer than the multiple dozens of DOFs of the hand.

The aim of the present study is to assess whether hand movements exist in a manifold whose dimensionality is lower than its maximum value, defined by the number of DOFs. To this end, we have human participants perform two manual behaviors —grasping and signing in American Sign Language (ASL)—while we track their hand kinematics. We then assess the degree to which low-variance PCs are structured by quantifying the degree to which they carry information about the manual behavior. For example, humans and nonhuman primates precisely preshape their hands when grasping objects. Objects can thus be classified on the basis of the hand postures adopted while grasping them, even before contact. If low-variance PCs indeed reflect noise, they should bear no systematic relationship with the object to be grasped. If low-variance PCs reflect subtle but volitionally controlled adjustments of hand posture to better preshape to the object, these should also be highly object specific.

## Results

**Basic structure of hand kinematics for two manual tasks.** Subjects performed two tasks—grasping various objects and signing in ASL—while we tracked their hand movements using a camera-based motion tracking system (Fig. 1). First, we examined the degree to which the structure of hand movements differed across tasks and individuals. Second, we ascertained the degree to which actual hand movements occupy a low-dimensional manifold of the available space spanned by the hand's DOFs.

As might be expected, different joint trajectories were observed when subjects grasped different objects or signed different ASL letters (Fig. 2). Moreover, the kinematics were consistent within condition—grasping a specific object or signing a specific letter—as

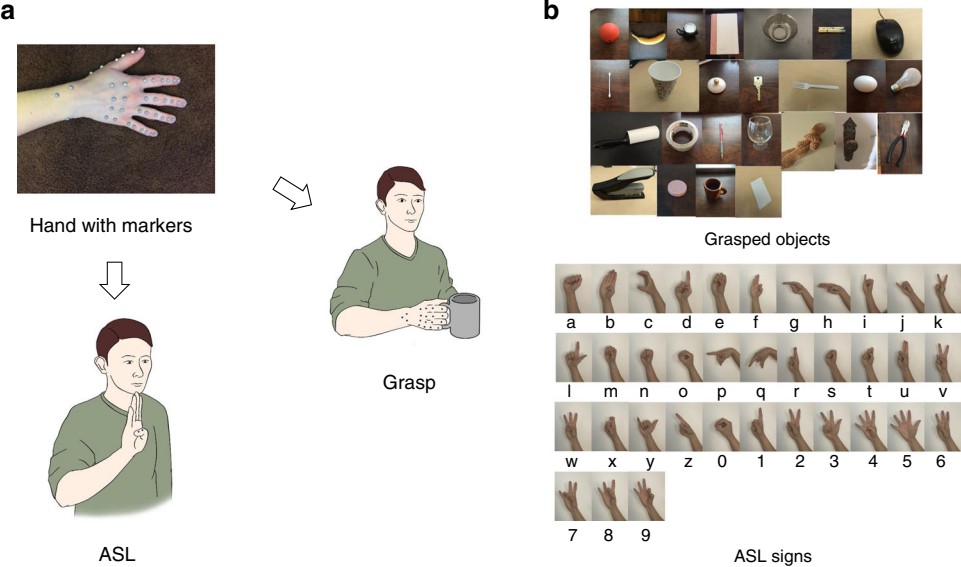

**Fig. 1 Experimental design. a** Subjects performed two manual tasks: grasping objects and signing in American Sign Language (ASL). Infrared cameras tracked the 3D trajectories of markers placed on the subjects' hand and joint angles were calculated from these marker trajectories using reverse kinematics. **b** Grasped objects and ASL signs.

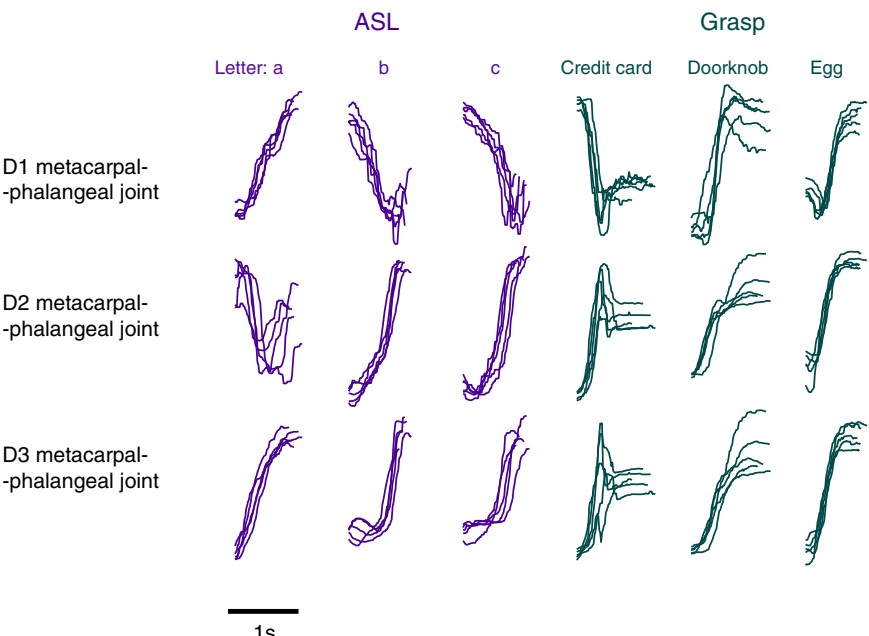

**Fig. 2 Example kinematic traces.** Joint kinematics (metacarpophalangeal joints flexion of digits 1–3) when grasping three different objects or signing three ASL signs five times each. Each trace shows a different trial of the same object/ASL sign.

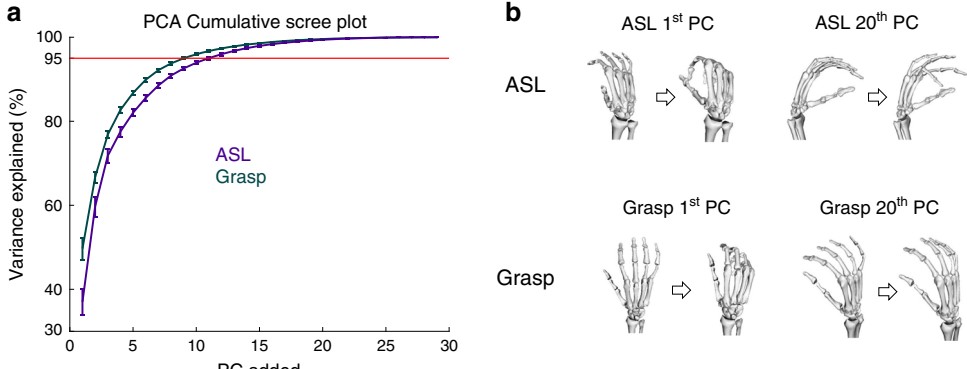

**Fig. 3 Principal components analysis. a** Cumulative percentage of variance explained vs. number of principal components. PCA is performed separately for each task and subject. The curves are averaged across eight subjects for grasp and across three subjects for ASL. Error bars denote the standard error of the mean. **b** Visualization of example PCs. For each task, we show the 1st and the 20th PCs (in terms of variance explained). The 20th PC accounts for less than one percent of the variance.

evidenced by similar trajectories over repetitions of the same condition (Fig. 2).

First, we wished to reproduce previous findings that much of hand kinematics can be described within a low-dimensional subspace. To this end, we performed PCA on the joint angles and examined the cumulative variance plot. We found that 3–5 PCs were sufficient to account for 80% of the variance in the kinematics and that 8–11 PCs accounted for 95% of variance, consistent with previous findings (Fig. 3a)[12,23]. We then examined PCs with large and small eigenvalues (Fig. 3b). In line with previous findings, the first two PCs of grasp and ASL involved opening and closing the hand, engaging mostly metacarpophalangeal (MCP) joint flexion/extension, some proximal and distal interphaleangeal joint flexion/extension (PIP and DIP), and some wrist flexion. One might expect PCs with small eigenvalues to reflect motor or measurement noise, and thus be unstructured. Instead, examination of low-variance PCs (for example, the 20th) revealed coordinated joint movements (e.g.,

ring PIP and MCP flexion in ASL) that were systematically dependent on condition (Fig. 3b).

We then compared the hand kinematics in the two tasks—grasping and ASL—by comparing their respective kinematics subspaces using cross-projection similarity. For each subject doing both tasks ($N = 3$), we calculated how much variance in the kinematics of one task were accounted for by the leading dimensions of the other (within-subject similarity). Despite the apparent dissimilarity of grasping and signing movements, we found that the underlying subspaces were very similar: The leading ten PCs of grasping explained ~85% of the variance in ASL kinematics and vice versa (Supplementary Fig. 1). We also found that different subjects yielded similar subspaces (Supplementary Fig. 1).

**Structure of low-variance PCs.** Having replicated previous results that hand movements can be reconstructed with high precision using a reduced basis set, and having shown that similar

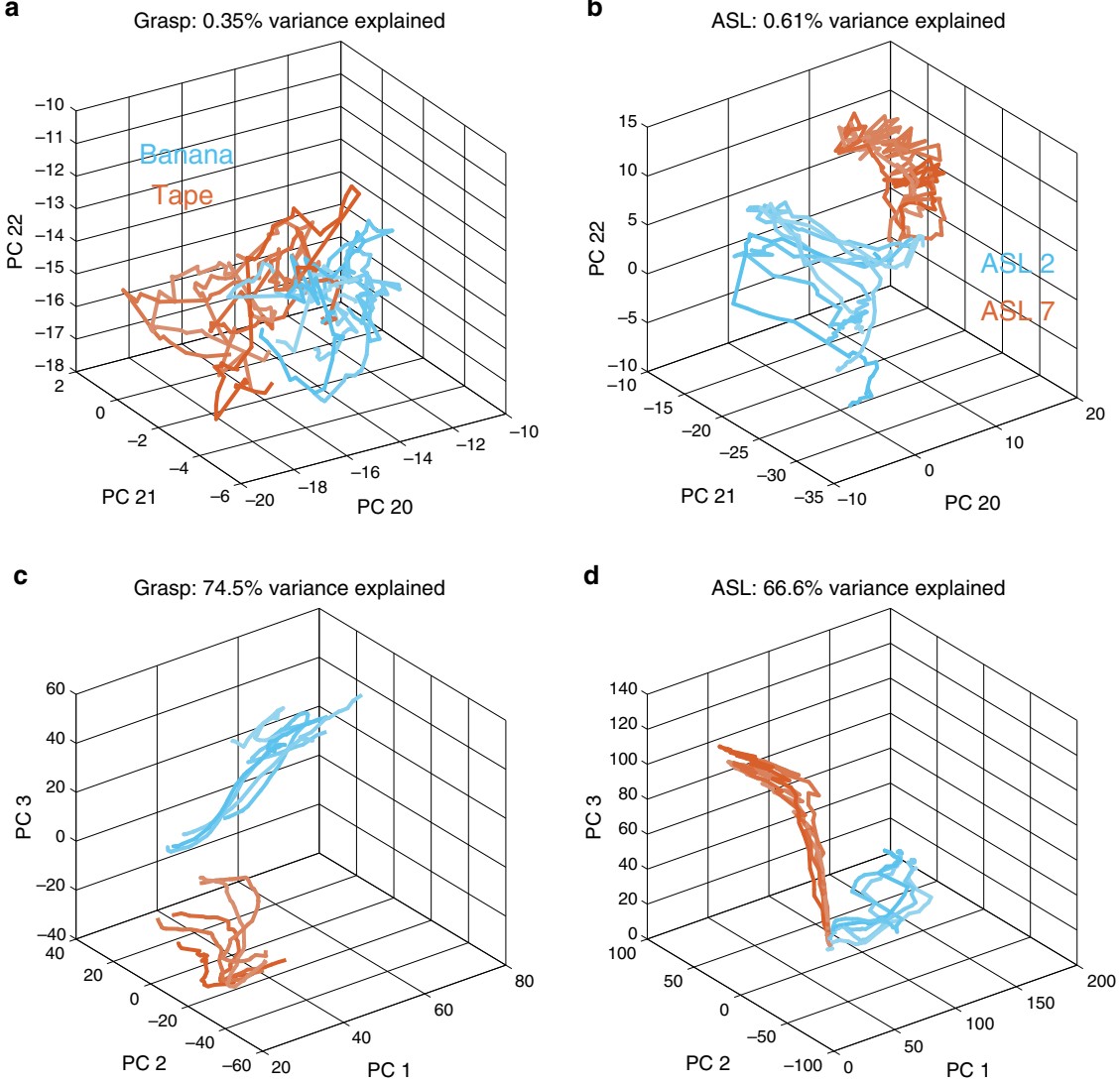

**Fig. 4 Object-specific kinematics in PC subspaces. a** Projection of the kinematics onto PCs 21 through 23 for five grasps each of tape and banana. **b** Projection of the kinematics onto PCs 21 through 23 for five repeated signs each of "two" and "seven" in ASL. **c** Projection of the kinematics onto PCs 1 through 3 for the conditions shown in **a**. **d** Projection of the kinematics onto PCs 1 through 3 for the conditions shown in **b**. In all panels, separate objects and signs are indicated by color, and different trials are indicated by traces of different lightness.

basis sets accounted for movements across both tasks and subjects, we then examined whether low-variance PCs outside these basis sets were structured in a condition-specific manner. We found that the kinematics projected on the low-variance PCs across repeated trials for the same condition (same grasped object, same ASL letter) varied systematically with the grasped object or signed letter (Fig. 4a, b), although this structure was noisier than that for the high-variance PCs (Fig. 4c, d), as might be expected. In other words, even the low-variance PCs reflect structure rather than only noise in the kinematics. Indeed, the trajectories along all PCs were far more consistent within conditions than they were across conditions (Supplementary Fig. 2). To quantify the degree to which kinematics differed across conditions, we classified objects or letters using progressively reduced kinematic subspaces that captured monotonically less variance through the systematic removal of PCs (in decreasing order of variance explained). We found that classification accuracy was well above chance even after most PCs had been removed, and that high performance was achieved even when all the remaining PCs accounted for less than one percent of the variance in kinematics (Fig. 5a, b). A similar result was obtained when

progressively removing linear discriminant analysis (LDA) dimensions rather than PC dimensions (Supplementary Fig. 3). Results from these classification analyses are thus inconsistent with the hypothesis that low-variance PCs reflect motor or measurement noise. Rather, these PCs seem to reflect subtle dimensions of movement that are under volitional control and contribute to the exquisitely precise preshaping of the hand to an object or to the detailed execution of a complex hand conformation required to produce an ASL sign.

**Nonlinear manifolds of hand kinematics.** We showed that low-variance PCs are structured and task related. However, given that PCA is a linear dimensionality reduction technique, we considered the possibility that hand kinematics occupy a low-dimensional nonlinear manifold, and that the low-variance PCs reflect a linear approximation of nonlinear dimensions. Indeed, such a nonlinearity could in principle explain why low-variance dimensions carry condition-specific information, thereby supporting the classification of object identity or ASL sign (Fig. 5). To address this possibility, we performed the same classification

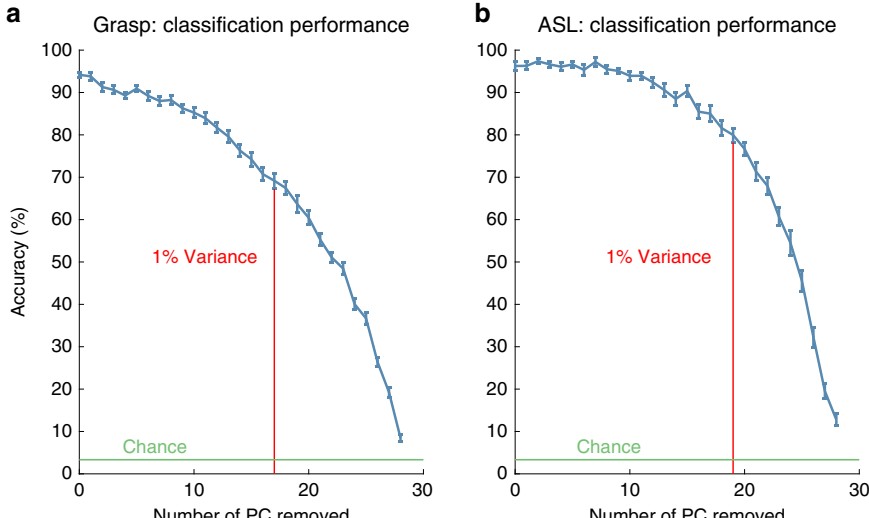

**Fig. 5 Classification of condition (object, sign) based on different subspaces of kinematics. a** Mean grasp classification performance after progressively removing PCs, from high variance to low. Results are averaged across eight subjects with five repetitions each (random allocation of training and testing data in each repetition, see "Methods" section for more details). Error bars denote the SEM. **b** Mean ASL classification performance based on reduced kinematic subspaces, averaged across three subjects with five repetitions each. Vertical red lines denote the PC beyond which the subspace accounts for <1% of the variance; error bars denote the standard error of the mean (SEM). Note that objects and letters can be classified accurately based on just a handful of high-variance PCs (Supplementary Fig. 4). Classification performance is highly consistent both across repetitions of the same subject and across subjects (note the small SEM).

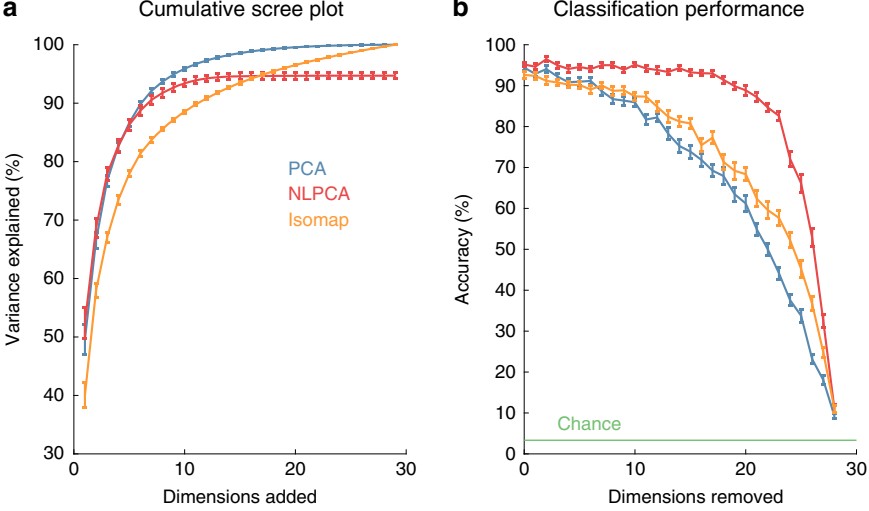

**Fig. 6 Scree plot and classification performance using different dimension reduction techniques. a** Cumulative variance explained by each dimension of PCA, NLPCA, and Isomap, averaged across the down-sampled data of eight subjects. **b** Mean grasp classification performance after progressively removing dimensions from high variance to low, for each of the three dimensionality reduction techniques. Results are averaged across eight subjects with five repetitions each. Error bars indicate ± SEM.

analysis with two nonlinear dimension reduction techniques, namely Isomap[24] and nonlinear PCA (NLPCA)[25]. Consistent with previous findings[26], we found that PCA provides the most parsimonious representation of the kinematics, as gauged by variance explained (Fig. 6a), a counterintuitive result given that PCA is restricted to linear transformations, whereas the other two approaches are not. More importantly, all three algorithms yielded high classification performance (>50%) after removing the 20 leading dimensions, which collectively account for >90% of the variance (Fig. 6b). If hand kinematics were low-dimensional and nonlinear, nonlinear dimensionality reduction would account for variance more parsimoniously and a smaller number of dimensions would contain all the task-relevant information, but this is not what we found. We conclude the high information content in

low-variance PCs is not a trivial artifact of nonlinearity, at least not of a nonlinearity that could be captured by the two well established approaches to nonlinear dimensionality reduction used here. The possibility remains that a low-dimensional manifold exists that cannot be captured with either Isomap or NLPCA.

**Conditional noise**. Next, we examined the possibility that our ability to classify objects based on low-variance PCs might be an artifact of condition-dependent noise. Indeed, the variability in the kinematics could in principle depend on which object is being grasped, as might be predicted by optimal feedback control[27]. This condition-dependent noise might then shunt signal to the

low-variance components to in turn be exploited by the classifier. To test this possibility, we simulated kinematics by (linearly) reducing the kinematics to ten dimensions and adding noise whose structure depended on the object being grasped. We then assessed the degree to which we could classify objects based on the low-variance PCs. We found that, with the addition of condition-dependent noise, the first ten PCs accounted for even less variance in the kinematics than they did in the original kinematics, despite the fact that, in principle, all the signal was confined to those dimensions (Supplementary Fig. 5). Nonetheless, classification performance with low-variance PCs (>10) was above chance with these simulated kinematics, reflecting the influence of condition-dependent noise (Supplementary Fig. 6c). However, performance dropped much faster as PCs were removed with the simulated low-dimensional kinematics than with the original kinematics (Supplementary Fig. 6c), despite the fact that low-variance PCs accounted for more variance in the former than in the latter. Furthermore, the correlation between within-object PC scores was near zero for the low-variance PCs (Supplementary Fig. 7). That is, low-variance PCs in the simulated kinematics were not nearly as structured as were the low-variance PCs in the measured kinematics. In summary, condition-dependent noise yields classification performance with low-variance PCs that is only slightly above chance, and cannot account for the observed structure in the low-variance PCs of grasping and ASL kinematics.

## Discussion

We found that the structure of the hand postures adopted in the two tasks—grasping and signing ASL—were virtually indistinguishable. Indeed, grasping and ASL—which each included ~30 distinct conditions—yielded subspaces that were no more different than were the kinematic subspaces of different subjects performing the same task (Supplementary Fig. 1). At first glance, this finding seems to be inconsistent with previous observations that kinematics are more similar within task and across subjects than across tasks within subject[11,28]. However, our analysis differs from its predecessors in two fundamental ways. First, we sampled the two tasks across many conditions (objects, signs) in contrast to the more restricted conditions—such as manipulating a credit card or flipping book pages—used in previous studies. The subspaces we computed thus individually reflect a greater breadth of possible hand conformations than would subspaces computed from much more limited tasks. Second, the two tasks that we implemented did not entail contact with objects, which introduces hand conformations that cannot be achieved without contact (for example, extension of the DIP). A manifold computed from kinematics before and after object contact would thus reflect not just volitionally achieved kinematics, but also the structure of the object. Systematic examination of other manual behaviors with and without object contact will be necessary to conclusively establish the degree to which contact shapes kinematics.

One hypothesis derived from optimal feedback control theory stipulates that the CNS defines low-dimensional manifolds of control to satisfy movement goals on a task-by-task basis, with motor noise being preferentially shunted into dimensions outside of such manifolds[27–29]. One possibility, then, is that the condition-specific information in the low-variance PCs reflects the influence of condition-dependent noise, which could in principle support classification of objects or ASL signs. However, kinematic trajectories projected onto low-variance PCs were far more consistent within than across conditions (Supplementary Fig. 2). Such consistency would not be present if these dimensions simply reflected motor noise shunted into those dimensions.

One might argue that nonlinear dimensionality reduction is better suited to reveal the dimensionality of hand postures. However, linear approaches, including PCA, have been previously shown to yield more efficient and reliable manifolds for kinematics than do nonlinear ones[26]. Not only do we replicate this result, but we also show that nonlinear dimensionality reduction does not capture behaviorally relevant aspects of the hand kinematics more efficiently than does PCA (Fig. 6). Overall, our results are consistent with the hypothesis that hand postures occupy a high-dimensional manifold, even for an everyday manual behavior, such as grasping.

High-dimensional kinematics do not imply wholly unconstrained control of the hand. Indeed, single-finger movements attempted by monkeys[30] and humans[31] are never perfectly individuated: they comprise incidental movements of the other digits arising in part from co-contraction of musculature associated with other digits. Recently, the analysis of the constraints imposed on neural activity in primary motor cortex (M1) revealed that activity patterns are constrained not only to a particular linear subspace[32], but also to a particular bounded region—a "repertoire"—within that subspace[33]. The known constraints imposed on volitional hand movements might be better explained in terms of a bounded repertoire within a high-dimensional subspace rather than an unbounded repertoire within a low-dimensional subspace.

The complexity of hand movements might be interpreted as evidence against the general notion that motor control occupies a low-dimensional manifold[34]. Another possibility, however, is that the hand is a uniquely complex effector and constitutes an exception to this rule. Indeed, the kinematics of the hand are higher dimensional than are those of other effectors[16,35–37]. Moreover, a subdivision of primate M1 has direct access to motoneurons that innervate muscles, particularly muscles of the hand[6], which could be construed as anatomical support for the notion that hand control is special. A common argument in support of low-dimensional motor control is that the brain needs to simplify the problem of controlling a complex effector, such as a hand, to solve it. Note, however, that control signals required for a 29 DOF effector still occupy a neural manifold whose dimensionality is much lower than that of the possible neural space spanned by the activity of all neurons modulated by the task[34]. Recent advances in large-scale neuronal recordings suggest that high-dimensional representations are possible if not common: sensory representations of natural scenes in primary visual cortex exceed 500 dimensions[38], an order of magnitude more than the implied representations of hand postures. Nonetheless, visual percepts are highly intuitive and allow for the accurate, rapid, and effortless identification of complex objects[39]. In comparison, motor control is positively straightforward!

## Methods

**Experimental design.** All procedures were approved by the Institutional Review Board of the University of Chicago. Eight right-handed adult subjects between the ages of 21–40 participated in the experiment with written consent. During the experiment, subjects were instructed to perform two types of manual tasks: grasping objects and signing in ASL (Fig. 1a). All eight subjects performed the grasping tasks and three subjects—who had prior knowledge of ASL—performed the signing task. All subjects performed one or both tasks with their dominant hand, the right one.

In the grasping task, subjects began each trial by resting their right hand on a table in front of them. The experimenter then placed an object at the center of the table and subjects grasped the object with their right hand, lifted it, and held it up for ~1 s before replacing the object on the table and moving their hand back to the starting position. No time limit was imposed on the trial and this procedure was repeated five times for each object. Twenty five objects, varying in size, shape, and orientation, were used to elicit 30 distinct grasps (Fig. 1b), with more grasps than objects arising from the fact that some objects could be grasped in several ways. For example, a lightbulb can be grasped by the stem or by the bulb. Objects that

afforded more than one grasp were presented repeatedly and subjects were cued to use a specific grasp on each presentation.

In the ASL task, subjects began at the same position as in the grasping task. On each trial, subjects signed an ASL sign—one of the 26 letters of the alphabet or a number from 1 to 10—and repeated it five times. Again, no time limit was imposed.

**Measurement and preprocessing.** Forty one infrared-reflective markers (hemispherical, 4-mm diameter) were placed on the right hand of each subject, with two markers covering each finger joint, two on the ulnar, and one on the radial bone of the forearm (Fig. 1a). Fourteen infrared cameras (8MP resolution, 250 Hz; MX-T Series, VICON, Los Angeles, CA) fixed to wall mounts and camera stands tracked the 3D trajectories of each marker (100-Hz sampling rate), each of which was then labeled based on its respective joint using Vicon Nexus Software (VICON, Los Angeles, CA). We then calculated inverse kinematics using time-varying marker positions and a musculoskeletal model of the human arm (https://simtk.org/projects/ulb_project)[40–46] implemented in Opensim (https://simtk.org/frs/index.php?group_id=91)[47]. The model was modified to include three rotational DOF of the first and fifth carpo-metacmatal joints to permit reconstructions of oppositional movements of these digits. In total, we reconstructed the time-varying angles of 29 DOF, including all movement parameters of the hand and three of the wrist. We only analyzed the intervals between the start of movement and 100 ms prior to object contact or until full ASL posture.

**PCs analysis and cross-projection similarity.** Kinematic synergies have been identified using PCA, which expresses hand postural trajectories in terms of a set of orthogonal components, each of which reflects correlated joint trajectories. We applied PCA to the hand kinematics obtained from each individual subject[48]. To compare PC subspaces across subjects or tasks, we computed the cross-projection similarity[11]. For this, we first calculated the total variance accounted for by the first N PCs of one group (V1). Then, we projected the kinematics from the first group onto the first N PCs of a second group (V2) and calculated the total variance explained. Finally, we computed the ratio V2/V1, which approaches 1 to the extent that the second subspace resembles the first. Note that this measure is not symmetric: if the first and second groups were to change roles in an alternative V2/V1 calculation, the resulting ratio would not necessarily be equivalent. Therefore, to obtain a symmetric similarity measure, we computed the ratio V2/V1 in both directions and report the average ratio as an index of subspace similarity.

**Classification.** Next, we assessed the degree to which hand kinematics were condition specific. That is, we quantified the extent to which hand postures were dependent on the object to be grasped or the letter/number to be signed. To this end, we used LDA to classify conditions based on the instantaneous hand posture (measured in joint angles) 100 ms before object contact, or when the ASL sign had been achieved. Classification performance with the full kinematics provided an upper bound on the achievable classification with LDA.

Then, we gradually removed PCs in descending order of variance and projected the hand posture of each trial based on a progressively smaller subset of non-leading PCs. We then used LDA on this restricted set to classify the grasped object or the ASL posture. We used a trial-level leave-one-out cross validation: for each object, we randomly select $M-1$ trials as training data (where $M$ is the total number of trials, $M = 5$) and trained a linear discriminant classifier on this training set. We then attempted to classify objects on the remaining trials. We repeated this procedure $M$ times (each with a different trial left out) and performance was quantified by the proportion of correct classifications.

**Nonlinear dimensionality reduction.** We used two nonlinear dimensionality reduction techniques to contrast with PCA: Isomap and NLPCA. We applied Isomap using the MATLAB package from Tenebaum et al.[24] with 29 nearest neighbors (though the results were robust to changes in this parameter over a range). We calculated the variance explained by each Isomap dimension by dividing the eigenvalue of that dimension by the sum of eigenvalues. We then performed the same classification analysis by removing Isomap dimensions in descending order of eigenvalue. We also applied NLPCA, an autoencoder-based approach, using the MATLAB package from Scholz et al.[25]. NLPCA orders the hidden nodes (termed "nonlinear PCs", or NLPCs) by variance explained and enforces a PCA-like structure on the low-dimensional embeddings[25]. To obtain a cumulative variance plot, we calculated the variance explained by dividing the variance of the NLPCA-reconstructed kinematics by the total variance. We performed the same classification analysis by progressively setting NLPC scores to zero prior to reconstruction of the kinematics, starting with the NLPC that accounted for the most variance and proceeding in order of decreasing explained variance. Due to the high computational cost of nonlinear algorithms, we down-sampled the kinematics (100–20 Hz) when performing the nonlinear analyses. As a comparison, we also analyzed the down-sampled kinematics using PCA and plotted the results alongside those of the nonlinear algorithms (Fig. 6). The results of PCA on the down-sampled kinematics (variance explained and classification performance; Fig. 6) are almost identical to those on the full sample (Figs. 3a and 5).

**Conditional noise.** One possibility is that noise, especially non-isotropic, condition-dependent noise as might be anticipated by optimal feedback control theory, might push information about grasped objects into low-variance PCs and thereby support classification with those PCs. To address this possibility, we denoised the kinematics, reduced their dimensionality, then added condition-dependent noise to the resulting kinematic trajectories. Specifically, we selected one trial from each object and replicated it four more times to obtain a kinematics set that contained no within-condition noise. We then reconstructed the (denoised) kinematics with only the first ten PCs. Next, we drew from a multivariate Gaussian distribution with zero mean and a condition-specific covariance matrix. Specifically, we randomly shuffled joint angle order and recalculated the covariance matrix of the denoised data, repeating this procedure for each object. This way, the within-object covariance (noise) was consistently of a similar magnitude as was the between-object covariance (signal), but differed in orientation across objects. Then, we rescaled the conditional noise such that, when added to the 10-D kinematic trajectories, classification performance was similar to that using the original kinematics (~95.5%). We then computed the PCs of these simulated kinematics and performed the same classification analysis described above with sequentially removed PCs. We repeated the procedures above five times, each time using a different seed to generate the conditional noise distributions (by reshuffling the joint angles and resampling from the resulting distributions).

We also examined the correlation of the scores along each PC across trials to assess the degree to which individual PCs exhibited repeatable structure: for both the raw kinematics and the 10-D simulated kinematics with conditional noise, we computed the mean correlation coefficient across trials on which the same object was presented, and across trials on which different objects were presented after projecting the kinematics onto individual PCs. Note that all the noise in the simulated kinematics was condition dependent, thereby maximizing the degree to which trial-by-trial variability in the kinematics might support classification performance.

**Reporting summary.** Further information on research design is available in the Nature Research Reporting Summary linked to this article.

## Data availability

The hand kinematics data recorded in this study has been deposited in the Bensmaia lab repository on Github (https://github.com/yyan-neuro/BensmaiaLab/tree/master/HandKinematics). We also provide the data underlying each figure as a Source data file (SourceData.zip). The data underlying Figs. 3a, 5a, b, 6a, b and Supplementary Figs. 1, 2a, b, 3a, b, 4a, b, 5, and 6a–c are provided in "Source Data.xlsx". The kinematics trace data underlying Figs. 2, 3b and 4a–d are provided as a MATLAB file "ASL and Grasp.mat". The data underlying Supplementary Fig. 7a–c are provided as "Supplementary Fig. 7. mat". A reporting summary for this article is available as a Supplementary Information file. Source data are provided with this paper.

## Code availability

The custom Matlab code used in this study has also been deposited in the Bensmaia lab repository (https://github.com/yyan-neuro/BensmaiaLab/tree/master/HandKinematics). Source data are provided with this paper.

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

## Acknowledgements

We would like to thank Matthew Kaufman for help with the title. This work was supported by NINDS grant R01 NS082865.

## Author contributions

Y.Y., D.D.M., J.M.G., and S.J.B. designed the experiments. Y.Y., D.D.M., and J.M.G. collected the data. Y.Y. and J.M.G. analyzed the data. Y.Y., J.M.G., S.A.S., and S.J.B. wrote the manuscript.

## Competing interests

The authors declare no competing interests.
