## [Peer Review File · Nature Communications]

Reviewers' comments:

Reviewer #1 (Remarks to the Author):

Yan and collaborators investigate whether the control of hand postures is confined to a small subspace of the high-dimensional hand joint angle space. They argue that, if measured hand movements falling outside the subspace defined by the first few principal components (PCs) reflect motor or measurement noise, they should not be informative about the hand posture. Indeed, they show with a linear discrimination analysis (LDA) on the projection of the hand posture on a reduced set of PCs that postural dimensions that fall outside the subspace spanned by PCs explaining most of the variance are highly structured and informative about the hand posture. They conclude that volitional hand control occupies a high dimensional space.

The issue of whether low dimensionality in the observed kinematic patterns as captured by dimensionality reduction techniques such as PCA is a real feature of the neural control of hand movement is important. Thus, the conclusions of this work might have a significant impact in the field of motor control. However, it is critical to rule out all reasonable alternative explanations of the results compatible with the null hypothesis that volitional hand control is confined to a low-dimensional subspace.

One main concern is that variations in postural dimensions outside the low-dimensional subspace under volitional control may be informative even if due to noise. Indeed, it may critically depend on the noise characteristics. If noise is Gaussian and isotropic, it is reasonable to assume that it would not be informative. However, noise may be non-Gaussian, non-isotropic, signal dependent, and posture dependent. If noise differs across posture, it may be informative about the posture. For example, a simple simulation (see attached figure, mean and SD over 20 simulation runs of classification score with leave-one-out cross-validation on the top and variance explained on the bottom) in which 29-dimensional data is generated by selecting 30 random posture in a random 10-dimensional subspace (5 repetitions for each posture) and both posture dependent noise (Gaussian in a different random direction for each posture) and Gaussian isotropic noise are added, shows that PCs of order much higher than the dimensionality of the subspace can be highly informative. Rather than suggesting a specific model to explain the data presented in the manuscript, this simulation indicates that dimensions that fall outside the controlled subspace may be informative even if due to noise. Moreover, it raises a substantial skepticism about the feasibility of the proposed approach. Indeed, as it may be very difficult to identify the noise characteristics and to model its potential effect on the structure and information content of higher-order PCs, the fact that noise dimensions may be informative indicates that LDA or other classification approaches may not be suitable to test for the dimensionality of the controlled subspace.

A second concern is whether a non-linear low-dimensional model of neural control of hand movement might explain the results. If volitional control is achieved in a low-dimensional (linear) subspace but such space is mapped through a non-linear function onto posture space, the reconstruction of the original

space by PCA would be inaccurate and postural dimensions that fall outside the subspace spanned by the PCs explaining most of the variance might be informative because of the non-linearity of the underlying control manifold.

Andrea d'Avella

Reviewer #2 (Remarks to the Author):

The authors show that when performing a dimensionality reduction technique (PCA) on kinematics of hand postures during three tasks, the higher dimensions that explain lower amounts of variance nonetheless convey information about task conditions. When those higher dimensions are included in classification algorithms, performance improves. The authors suggest that this contradicts standard interpretations of postural synergies for the hand.

In general, I'm sympathetic to the authors' point of view and definitive studies on this topic would be welcome and disproving this hypothesis would be significant. However, I don't think this study makes significant contribution to the topic. I went back to the early Santello et al 1998 paper that started the idea of kinematic hand synergies and found that very similar analyses and results were reported there. From the abstract: "However, even though they were small, higher-order (more than three) principal components did not represent random variability but instead provided additional information about the object." They performed a similar type of information analysis as performed in the present study to reach this conclusion (Fig 9). There are some subtle differences in the results (i.e. they kind of eyeball their plots and say that 6 components are good enough; but, as discussed below, they don't conclude that there are therefore 6 synergies and everything else is noise), but the main idea that low variance components convey information is clearly stated in that paper. It's therefore not clear to me how the current study advances the field over what has previously been published.

Moreover, the synergy hypothesis stated in this manuscript - that higher order components represent noise - is something of a simplified straw-man. Again, from the Discussion of Santello et al., they were careful to say that they are not arguing for that point of view. They suggest that their results imply a 'two level' control scheme, in which a low number of basic synergies might be used to crudely control the hand and get it in useful basic postures but that fine adjustments can be made to adapt hand postures to the particular task. In this interpretation, the higher dimensional components are not random noise but represent task relevant alterations in movements that act on top of the basic postural synergies. Similar suggestions can be found in other studies and perspectives on synergies. While one can debate whether this interpretation is actually useful (can it be disproven?), I don't think the results of the current study refute it.

One could argue that these discussion points have been obscured over the subsequent 20 years since that paper was published and that there is therefore a need to remind people of them. But that might be better done in a perspective piece or review article in a more specialized journal.

We would like to thank the reviewers for their thoughtful comments and have worked hard to address them, as detailed below. In summary: First, we implemented a simulation of low-dimensional kinematics to address the possibility that our classification performance was an artifact of condition-dependent noise. Second, we enlisted the help of Sara Solla – an expert on non-linear manifolds and now a co-author on the paper – to guide us through the implementation of non-linear dimensionality reduction to the kinematics. Third, we discuss how our results constitute a fundamental departure from Santello's findings and articulate why they will have an important impact on the field. Finally, we have eliminated all analysis and discussion of the typing task because the hand kinematics measured in this task are contaminated by object contact, unlike those in the other two tasks. This review process has been remarkably constructive and we have the reviewers to thank for that.

Reviewer #1

Yan and collaborators investigate whether the control of hand postures is confined to a small subspace of the high-dimensional hand joint angle space. They argue that, if measured hand movements falling outside the subspace defined by the first few principal components (PCs) reflect motor or measurement noise, they should not be informative about the hand posture. Indeed, they show with a linear discrimination analysis (LDA) on the projection of the hand posture on a reduced set of PCs that postural dimensions that fall outside the subspace spanned by PCs explaining most of the variance are highly structured and informative about the hand posture. They conclude that volitional hand control occupies a high dimensional space.

The issue of whether low dimensionality in the observed kinematic patterns as captured by dimensionality reduction techniques such as PCA is a real feature of the neural control of hand movement is important. Thus, the conclusions of this work might have a significant impact in the field of motor control. However, it is critical to rule out all reasonable alternative explanations of the results compatible with the null hypothesis that volitional hand control is confined to a low-dimensional subspace.

One main concern is that variations in postural dimensions outside the low-dimensional subspace under volitional control may be informative even if due to noise. Indeed, it may critically depend on the noise characteristics. If noise is Gaussian and isotropic, it is reasonable to assume that it would not be informative. However, noise may be non-Gaussian, non-isotropic, signal dependent, and posture dependent. If noise differs across posture, it may be informative about the posture. For example, a simple simulation (see attached figure, mean and SD over 20 simulation runs of classification score with leave-one-out cross-validation on the top and variance explained on the bottom) in which 29-dimensional data is generated by selecting 30 random posture in a random 10-dimensional subspace (5 repetitions for each posture) and both posture dependent noise (Gaussian in a different random direction for each posture) and Gaussian isotropic noise are added, shows that PCs of order much higher than the dimensionality of the subspace can be highly informative.

Rather than suggesting a specific model to explain the data presented in the manuscript, this simulation indicates that dimensions that fall outside the controlled subspace may be informative even if due to noise. Moreover, it raises a substantial skepticism about the feasibility of the proposed approach. Indeed, as it may be very difficult to identify the noise characteristics and to model its potential effect on the structure and information content of higher-order PCs, the fact that noise dimensions may be

informative indicates that LDA or other classification approaches may not be suitable to test for the dimensionality of the controlled subspace.

We were able to replicate the result, put forth by Reviewer 1, that condition-dependent noise could support classification with low-variance principal components. We implemented the conditional noise differently than did the Reviewer, however, by drawing from a Gaussian distribution whose parameters are unique for each object. In the simulation, we only included condition-dependent noise (rather than a combination of both condition-dependent and condition-independent noise) to stack the odds against us, so to speak. That is, all of the variability in the kinematics is object specific and thus can in principle support classification performance.

Specifically, we first generated a 10-dimensional, denoised dataset from our original kinematics. We did this by randomly selecting one trial from each object and replace all other trials with it. We then reconstructed the data with only the first 10 PCs. Then, for every time point, we drew from a multivariate Gaussian with zero mean and a covariance matrix that was computed from the kinematics with joint angles randomly shuffled. This covariance matrix of shuffled joints was computed for each object and determined the noise applied to every time point of each trial for that object. That way, the noise had a similar covariance to the measured kinematics but differed across objects. We then scaled the conditional noise such that, when added to the 10-D denoised data, the classification performance with the full signal (i.e., without removing any PCs) was similar to that with the full measured kinematics signal (95%). We then examined classification performance and the correlation of kinematics trajectories projected onto PCs (**Supplementary Figure 6,7**). We also implemented the Reviewer's formulation of conditional noise for comparison (see below).

We found that, while the 10-D kinematics + conditional noise achieves above-chance performance for PCs 11 to 25, performance was far poorer than that achieved with measured kinematics despite the fact that all the noise was condition dependent (**Supplementary Figure 6C**). Furthermore, the correlations between same-object trajectories and between different-object trajectories, projected onto individual PCs, were very different for the simulated and measured kinematics (**Supplementary Figure 7**). For the measured kinematics, the mean Pearson correlation of same-object trials was consistently much higher than that for different-object trials (**Supplementary Figure 7A**). For the simulated kinematics, the same-object and different-object correlations are both near zero for low-variance PCs (**Supplementary Figure 7C**). Thus, low-variance PCs in the simulated (low-dimensional) data do not exhibit the structure observed in their measured counterparts. Even with Reviewer 1's implementation of conditional noise, we obtain weaker classification performance (Permutation ANOVA, $p < 1e-10$, $n=6930$, permutation number = 1000) and unstructured low-variance PCs (see **Figure 1 A, B below**) compared to their counterparts computed from measured kinematics.

In summary, the addition of conditional noise to low-dimensional simulated kinematics somewhat supports classification with low-variance PCs but does not feature the structure in these PCs that we observe in the measured kinematics.

A second concern is whether a non-linear low-dimensional model of neural control of hand movement might explain the results. If volitional control is achieved in a low-dimensional (linear) subspace but such space is mapped through a non-linear function onto posture space, the reconstruction of the original space by PCA would be inaccurate and postural dimensions that fall outside the subspace spanned by

the PCs explaining most of the variance might be informative because of the non-linearity of the underlying control manifold.

This is an excellent point. To address it, we enlisted the help of a world expert on non-linear manifolds to guide us through our attempts to address it. One challenge is that most non-linear dimensionality reduction approaches (e.g. t-SNE, autoencoder, LLE) do not enforce an orderly structure in estimating dimensionality, making it difficult to isolate low-variance dimensions and assess their informativeness. With this in mind, we selected two non-linear techniques that are not subject to this limitation, namely Isomap¹ and non-linear PCA (NLPCA)². Isomap dimensions are ordered by eigenvalues similar to PCA [1], while NLPCA is an autoencoder-based approach that arranges the hidden neurons in order of variance explained [2]. Isomap has been shown to be quite effective in identifying global non-Euclidian structure^{1,3} and NLPCA can also accommodate a wide range of non-linearities².

If, as the reviewer points out, kinematics occupied a low-dimensional non-linear manifold, we would expect classification performance to fall to near chance after these dimensions were removed. Instead, we find that all three algorithms yield high performance (50% ~ 80% accuracy for 30 objects) even when the top 10 to 20 dimensions are removed (**Figure 7B**). In fact, performance is similar or even better than that of PCA. That is, the low-variance components in the non-linear manifold carry even more information about objects than do their linear counterparts. We conclude that the high dimensionality we observe is not a trivial consequence of a non-linearity in the kinematics, at least not in a non-linearity that can be captured using these two non-linear dimensionality reduction approaches.

Note that PCA outperforms both NLPCA and Isomap in terms of parsimony of the description of the kinematics (**Figure 7A**). Fewer dimensions are required to explain the data with PCA than with Isomap and NLPCA. This is counterintuitive because PCA is linear and the other two are not, but this counterintuitive result has been previously documented⁴.

Figure 1. A | Classification performance as PCs are removed. We used the 10-dimensional de-noised data + Reviewer 1's conditional noise. B | Correlation between trials from the same object or different objects onto different PCs. We used the same dataset as in A.

Reviewer #2

The authors show that when performing a dimensionality reduction technique (PCA) on kinematics of hand postures during three tasks, the higher dimensions that explain lower amounts of variance nonetheless convey information about task conditions. When those higher dimensions are included in classification algorithms, performance improves. The authors suggest that this contradicts standard interpretations of postural synergies for the hand.

In general, I'm sympathetic to the authors' point of view and definitive studies on this topic would be welcome and disproving this hypothesis would be significant. However, I don't think this study makes significant contribution to the topic. I went back to the early Santello et al 1998 paper that started the idea of kinematic hand synergies and found that very similar analyses and results were reported there. From the abstract: "However, even though they were small, higher-order (more than three) principal components did not represent random variability but instead provided additional information about the object." They performed a similar type of information analysis as performed in the present study to reach this conclusion (Fig 9).

There are some subtle differences in the results (i.e. they kind of eyeball their plots and say that 6 components are good enough; but, as discussed below, they don't conclude that there are therefore 6 synergies and everything else is noise), but the main idea that low variance components convey information is clearly stated in that paper. It's therefore not clear to me how the current study advances the field over what has previously been published.

First, Santello 1998 tracked half as many joints as we did, which thus limits the potential dimensionality of the resulting kinematics. He found that 2 PCs accounted for 80 to 87% of the kinematics variance and labelled PCs 3 and beyond as "higher-order PCs." Figure 9 in Santello 1998 shows that cumulative information (about grasped object) still increases for PC 3-6 but plateaus afterwards, so they concluded that "higher order PCs (PC 3-6) contain information and the degree of freedoms of human hand is at least 5 to 6"⁵. From our perspective, this is still very low dimensional: we need 8 to 10 PCs to explain 95% data variance and our stated dimensionality is more like 20 to 25.

Second, the ostensible conclusion from this line of work – from the citation referenced above and dozens of others – is that hand postures occupy a lower dimensional manifold than is afforded by hand biomechanics. That one sentence in Santello 1998 may open the door for higher dimensionality, but that is certainly not the conclusion from that or the other papers.

Moreover, the synergy hypothesis stated in this manuscript - that higher order components represent noise - is something of a simplified straw-man. Again, from the Discussion of Santello et al., they were careful to say that they are not arguing for that point of view. They suggest that their results imply a 'two level' control scheme, in which a low number of basic synergies might be used to crudely control the hand and get it in useful basic postures but that fine adjustments can be made to adapt hand postures to the particular task. In this interpretation, the higher dimensional components are not random noise but represent task relevant alterations in movements that act on top of the basic postural synergies. Similar suggestions can be found in other studies and perspectives on synergies. While one can debate whether this interpretation is actually useful (can it be disproven?), I don't think the results of the current study refute it.

One could argue that these discussion points have been obscured over the subsequent 20 years since that paper was published and that there is therefore a need to remind people of them. But that might be better done in a perspective piece or review article in a more specialized journal.

It is true that Santello et al. 1998 does not explicitly argue that higher order components reflect noise. However, the general consensus in the motor control community, arising from this and other papers, is that volitional hand movements exist in a low-dimensional manifold. This idea is ubiquitous in studies of manual behavior, is not explicitly contradicted in any study that we are aware of, and necessarily implies that dimensions of movement falling outside this putative manifold are dominated by noise. Moreover, even those who maintain that volitional control of the hand is higher-dimensional than typically reported will very likely be keenly interested in precisely how high-dimensional the manifold is. We report that more than twenty dimensions of hand kinematics show signs of being under volitional control, at least twofold more than is typically reported. As a result, we feel this demonstration of the structure in high-variance PCs will have a significant impact on the motor control community.

References

1. Tenenbaum, J. B., De Silva, V. & Langford, J. C. A global geometric framework for nonlinear dimensionality reduction. *Science (80-.)*. **290**, 2319–2323 (2000).
2. Scholz, M., Fraunholz, M. & Selbig, J. Nonlinear principal component analysis: Neural network models and applications. *Lect. Notes Comput. Sci. Eng.* **58**, 44–67 (2008).
3. Van Der Maaten, L. J. P., Postma, E. O. & Van Den Herik, H. J. Dimensionality Reduction: A Comparative Review. *J. Mach. Learn. Res.* **10**, 1–41 (2009).
4. Patel, V., Burns, M., Mao, Z.-H., Crone, N. E. & Vinjamuri, R. Linear and nonlinear kinematic synergies in the grasping hand. *J. Bioeng. Biomed. Sci.* (2015).
5. Santello, M., Flanders, M. & Soechting, J. F. Postural Hand Synergies for Tool Use. *J. Neurosci.* **18**, 10105–10115 (1998).

****REVIEWERS' COMMENTS:**

Reviewer #1 (Remarks to the Author):

The authors have fully addressed my comments by performing additional analyses.

Reviewer #2 (Remarks to the Author):

I don't have major concerns with the quality of the work in the manuscript or the basic results presented. I still have the concerns about the novelty and significance of this work that I expressed in the previous round. I would suggest that, at a minimum, the manuscript should be revised so that the links of the current study to that previous work are clear. The idea that higher dimensions were informative - the main point of the current study - was not just expressed in a single sentence in Santello et al., but was clearly stated and analyzed. This should be clear in the Introduction and Discussion of the current paper to better reflect the novelty and significance of the current work.

I like the addition of the nonlinear dimensionality analyses. However, I'm not sure whether this fully addresses the basic issue about whether the current study excludes the possibility that there could a lower dimensional control strategy expressed at the level of muscles (or neurons). Given the complex relationship between muscle activation and hand kinematics, it seems entirely possible that an apparent high dimensional (linear or nonlinear) kinematics could be generated from low dimensional muscle control commands. I don't think that even the nonlinear analyses examined here would necessarily reveal that lower dimensional control space, if it existed. This possibility and limitation should be clearly stated in the Discussion.

We would like to thank the reviewers for their helpful comments. We have made additional changes to the manuscript to address them, as detailed below. We have also made some minor clarifying edits throughout the document to improve legibility.

I don't have major concerns with the quality of the work in the manuscript or the basic results presented. I still have the concerns about the novelty and significance of this work that I expressed in the previous round. I would suggest that, at a minimum, the manuscript should be revised so that the links of the current study to that previous work are clear. The idea that higher dimensions were informative - the main point of the current study - was not just expressed in a single sentence in Santello et al., but was clearly stated and analyzed. This should be clear in the Introduction and Discussion of the current paper to better reflect the novelty and significance of the current work.

We have modified the following passage to frame the question in the context of previous work:

"The tacit assumption underlying the interpretation of this dimensionality reduction is that high-variance principal components – the synergies – are under volitional control whereas low-variance PCs reflect motor or measurement noise. Another possibility, however, is that the exquisite control of the hand is mediated by high dimensional sensorimotor signals, and that low-variance PCs are critical to achieving precise hand postures. A previous investigation of this question concluded that grasping movements were confined to 6 dimensions or less²²— still far fewer than the multiple dozens of DOFs of the hand."

Reference 22 is the Santello paper in question.

I like the addition of the nonlinear dimensionality analyses. However, I'm not sure whether this fully addresses the basic issue about whether the current study excludes the possibility that there could be a lower dimensional control strategy expressed at the level of muscles (or neurons). Given the complex relationship between muscle activation and hand kinematics, it seems entirely possible that an apparent high dimensional (linear or nonlinear) kinematics could be generated from low dimensional muscle control commands. I don't think that even the nonlinear analyses examined here would necessarily reveal that lower dimensional control space, if it existed. This possibility and limitation should be clearly stated in the Discussion.

We have expanded the following passages to acknowledge this possibility:

Results section:

"We conclude the high information content in low-variance PCs is not a trivial artifact of non-linearity, at least not of a non-linearity that could be captured by the two well established approaches to non-linear dimensionality reduction used here. The possibility remains that a low-dimensional manifold exists that cannot be captured with either Isomap or NLPCA."

To the extent that kinematics are the product of muscle activations, kinematics cannot be higher dimensional than muscle activations from the perspective of volitionally controlled degrees of freedom. Any non-linearity in the kinematics manifold should be captured by the non-linear dimensionality reduction analysis, with the caveat stated in the above passage.